# A Lightweight Multi-Source Fast Android Malware Detection Model

**Tao Peng** 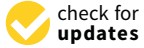**, Bochao Hu** **, Junping Liu \*, Junjie Huang, Zili Zhang, Ruhan He and Xinrong Hu**

School of Computer and Artificial Intelligence, Wuhan Textile University, No. 1 Sunshine Avenue,
Jiangxia District, Wuhan 430200, China; pt@wtu.edu.cn (T.P.); hubochao007@gmail.com (B.H.);
jjhuang@wtu.edu.cn (J.H.); zlzhang@wtu.edu.cn (Z.Z.); heruhan@wtu.edu.cn (R.H.); hxr@wtu.edu.cn (X.H.)
\* Correspondence: jpliu@wtu.edu.cn

**Abstract:** Most of the current malware detection methods running on Android are based on signature and cloud technologies leading to poor protection against new types of malware. Deep learning techniques take Android malware detection to a new level. Still, most deep learning-based Android malware detection methods are too inefficient or even unworkable on Android devices due to their high resource consumption. Therefore, this paper proposes MSFDroid, a lightweight multi-source fast Android malware detection model, which uses information from the internal files of the Android application package in several dimensions to build base models for ensemble learning. Meanwhile, this paper proposes an adaptive soft voting method by dynamically adjusting the weights of each base model to overcome the noise generated by traditional soft voting and thus improves the performance. It also proposes adaptive shrinkage convolutional unit that can dynamically adjust the convolutional kernel's weight and the activation function's threshold to improve the expressiveness of the CNN. The proposed method is tested on public datasets and on several real devices. The experimental results show that it achieves a better trade-off between performance and efficiency by significantly improving the detection speed while achieving a comparable performance compared to other deep learning methods.

**Keywords:** android malware dectection; multi-source; lightweight model; deep learning; ensemble learning



## 1. Introduction

### 1.1. Background

According to the "2020 Android Platform Security Situation Analysis Report" released by Qi'anxin Threat Intelligence Center [1], Qi'anxin Threat Intelligence Center intercepted a total of 2.3 million new malicious program samples on the Android platform in 2020 with an average of 6,301 new malicious program samples intercepted every day. Among them, the malicious deduction category accounted for 34.9%, the toll consumption category accounted for 24.2%, the rogue behaviour category accounted for 22.8%, the privacy theft category accounted for 12.3%, the lure fraud category accounted for 4.3%, and the remote control category accounted for 1.5%. From a global perspective, the security governance of the mobile Internet is relatively weak, especially the Internet banking theft Trojan horse virus is still widespread, showing a wide variety of types, methods and other characteristics, the threat to user property is serious. Internet banking Trojan horse virus often disguised as other applications to lure users into downloading and installing. Monitoring shows that Chrome (23.7%), Sagawa Express (8.2%, a famous Japanese courier application) and Flash Player (4.7%) are the applications with the highest number of counterfeits. Analysis of the attack techniques revealed that hacker groups mainly stole users' bank card credential information in the following four ways: first, using phishing pages, such as pop-up bank card binding pages to trick users into entering their bank card credentials; second, spoofing bank APPs, disguising Trojan horse programs as legitimate Internet banking APPs to steal

users' Internet banking information; third, pop-up phishing pages to cover bank APPs, where the attacker hides the Trojan horse program in the background of the phone. The Trojan hides in the background of the cell phone, and once the Internet banking starts, a phishing page will pop up to cover the original Internet banking page to trick users into entering their bank card credentials for theft; fourth, using accessibility services, after the Trojan starts, the user is asked to open the accessibility services provided by Android for people with disabilities to listen to the user's use of the banking APP, and the Trojan also records keyboard input information to steal bank card credentials. The Trojan also records keystrokes to steal bank card credentials. With the emergence of the Internet of Things, more and more IoT devices are equipped with Android. IoT devices' overall security protection and security management ability is far less than smartphones. Therefore, IoT devices have become a new target for many blackmail gangs. The old mining family AdbMiner is more active in targeting IoT devices. The Trojan continues to infect insecure IoT devices through specific ports to implement mining for revenue, so the security research for the Android system must consider IoT devices.

### 1.2. Motivation

Android has become the most popular operating system for smart mobile devices since its release in 2008. According to Statista, Android retained its position as the world's leading mobile operating system in June 2021, controlling the mobile operating system market with a 73% share. Google's Android and Apple's iOS together have more than 99 percent of the global market share [2]. However, due to various factors such as the open ecological model of Android, coarse-grained permission management, and the ability to call third-party code, many security attack surfaces have emerged, which seriously threaten the security of Android [3]. At present, while most of the current malware detection services for the Android platform need to be supported by cloud technology and a vast virus database established by authoritative security agencies, and such solutions are mostly on server side for app markets. In addition to the official Google Play, a large number of users will use third-party application markets such as Amazon App Store, GetJar, Mobogenie Market, etc., and these different application markets have different censorship of the applications on the shelves, when a new family of Android malware is reported, not all app marketplaces are able to respond within the response time. At the same time, there are also some users will choose to download applications from some third-party websites, the security of which cannot be guaranteed. Fingerprint databases that rely on hashing and cloud technologies are inefficient in coping with the huge number of new applications generated every day, and as with traditional methods such as signature-based malware detection (based on identifying specific patterns of known malware), malware can easily change its fingerprints to bypass such detection methods [4]. Instead of relying on a fixed fingerprint, artificial intelligence based technology Android malware detection method use machine learning or deep learning algorithms to automatically extract the most appropriate features and combinations of features to determine whether an Android application is a malware according to a pre-designed objective function. A large number of current research on Android malware detection based on artificial intelligence technology focuses on accuracy, and various complex models are designed to achieve accurate detection, but these models are too complex to achieve efficient detection on the user's Android devices. Currently, most malware detection tools running on the Android platform are implemented by comparing signatures through cloud technology or by uploading software installation packages to a server for detection, however, the traditional server-side based malware detection surely has unignorable drawbacks when detecting such apps, because (1) it is a time-consuming task to upload the apps to server before the installation, especially for large apps; (2) the uploading process via the Internet is not secure. For example, attackers may modify the malware during the uploading period such that an incorrect "benign" result is returned [5]. Therefore, a last line of defense on mobile devices is necessary and much-needed, it is necessary to propose a method that does not rely on fingerprints or

cloud technologies but rather an offline algorithm to efficiently discriminate malware before it is installed and run.

*1.3. Our Works*

To solve the above problems, we designed a lightweight and fast machine learning model. We used a computational server to complete the training of the model, and finally deployed the trained model to Android mobile devices. Experimental results showed that our model achieved efficiency while achieving considerable accuracy. In summary, our main contributions are as follows.

- We propose a multi-source approach for Android malware detection which uses multiple files in an Android application package. It extracts relevant features contained in the files from multiple dimensions, such as information in the file headers and the power spectral density of the structural entropy of the executable file, which makes the extraction of features more comprehensive.
- We studied the information in each field of the header of DEX files and found some key features in the DEX file header that can be used for malware detection. Therefore, a DEX header parser is proposed to extract key features from the DEX file header.
- We propose an adaptive shrinkage convolutional neural network, which can dynamically adjust the convolutional kernel weights and activation function thresholds through an attention mechanism, making the convolutional network with denoising ability while improving the expressiveness of the neural network model.
- We propose a new adaptive soft voting method, which can dynamically change the weight of each base model during the training process, overcoming the noise generated by the traditional soft voting due to the large performance gap and jitter of the base models, while significantly improving the performance of soft voting.

## 2. Related Work

The sandbox mechanism of Android makes it more challenging to monitor the dynamic behaviour of applications in non-custom systems. Many methods have been proposed for Android malware detection in many previous studies. Most of the traditional anti-virus techniques based on signature detection methods can detect known malware quickly and effectively, signature-based detection is mainly achieved by extracting signatures from malware and building malware libraries, but some malware can be hidden in the system by using different obfuscation and disguise techniques to the extent that it cannot detect unknown malware [6]. Machine learning algorithms usually have less than three layers of computational units and limited computational power to process raw data [7]. As a result, the performance of machine learning models relies heavily on the features extracted, and malware producers can bypass trained machine learning models by continuously updating their fraudulent techniques to harm users and companies. In the face of the increasing difficulty of Android malware detection, it is not easy to build a robust and transparent detection model or system through traditional machine learning techniques [8]. While deep learning is one of the mainstream algorithms in recent years, feature extraction by deep learning methods differs from conventional machine learning techniques. Deep learning can learn feature representations from the raw input data without requiring much prior knowledge. In addition, its ability to detect previously undetected types of malware based on identifying specific patterns of known malware can provide better performance in terms of detection efficiency and effectiveness, which is the key advantage of deep learning.

Techniques such as machine learning and deep learning are combined with program analysis techniques to infer the behavioral properties of applications. In the following, we will focus on two categories of program analysis techniques for the Android platform, static analysis and dynamic analysis, to discuss the related work and analyze the characteristics of these two categories of program analysis techniques.

## 2.1. Static Analysis

Static analysis is widely used for Android malware detection. The code is examined without execution, and the results are generated by analyzing the code structure, the sequence of statements, and how variable values are handled in different function calls. An example is the AndroidManifest.xml file, which describes the permissions, API calls, package names, referenced libraries, and application components. Another one is the classes.dex file contains all Android classes compiled into dex file format [9]. Some static methods can represent the analyzed application code as abstract models such as opcodes in the form of n-grams or other information about the program such as metadata (application description, application ratings, number of application downloads) depending on the purpose of the study, which can be collected from other perspectives for static analysis [3]. Daniel et al. [10] proposed Drebin, a lightweight mathod for Android malware detection that enabled identifying malicious applications directly on smartphone, Drebin performs a broad static analysis, gathering features and embedding them in a joint vector space, finally, the features were classified by support vector machines, achieving 94% accuracy. Zachariah et al. [11] looked at three aspects of static analysis (i.e., signature-based detection, permission-based detection and Dalvik bytecode detection), proposed methods for Android malware detection, and discussed the advantages and limitations of these methods. HybriDroid [12] extracted permissions, API calls, the number of users downloading the application and the rating of the application, and built a malware detection model using a nonlinear integrated decision tree forest (NDTF) approach with a detection rate of 98.8%. Xusheng et al. [13] used seven feature selection algorithms to select permissions, API calls, and opcodes, and then merged the results of each feature selection algorithm to obtain a new feature set. subsequently, they used this to train the base learner, and set the logical regression as a meta-classifier to learn the implicit information from the output of base learners and obtain the classification results and the F1-Score reached 96%. Although static analysis has some problems resisting malicious deformation techniques such as java reflection and dynamic code loading, static analysis is not only scalable and usable when facing batch unknown APKs detection, but also can traverse all possible execution paths of the APKs [14]. Moreover, static analysis can detect malware quickly and prohibit malware before installation, which is one of the key factors that will enable us to achieve our goals. so it is essential to use static analysis in Android malware detection.

## 2.2. Dynamic Analysis

Dynamic analysis techniques focus on runtime monitoring and profiling applications to obtain multiple behavioral characteristics and enable efficient malware detection. The dynamic analysis approach is used in a controlled environment to detect the application's behavior. Automatic dynamic analysis of Android applications requires user-simulated input event streams, such as touch, gestures, or clicks, to achieve more excellent code coverage when running in an emulator or on an actual phone [15]. The main objects of dynamic analysis include network traffic, battery usage, CPU utilization, IP addresses, and opcodes. One type of dynamic analysis relies on the Dalvik runtime or ART runtime to obtain the same level of privileges as the Android application, which usually requires modifications to the Android OS or the Dalvik virtual machine. Another type of dynamic analysis typically uses Android Virtual Devices (AVDs), emulators (Genymotion), or in real devices for data collection and analysis and achieves higher security through isolation [9]. Bläsing et al. [16] proposed a sophisticated kernel space sandbox that automatically executes applications without human interaction and saves system calls and logs. IntelliDroid [17] presented two input generation and injection techniques that iteratively detect event chains and compute the appropriate injected inputs and the injection order to enable them to trigger a broader code path. Dixon et al. [18] proposed a power-aware malware detection framework based on the method [19] that collects power samples and constructs power consumption based on the collected samples' history and generates success rate signatures based on the constructed history using noise filtering and data compression methods. The

Andromaly framework [20] proposed a dynamic feature-based classification framework. The framework consists of a host-based malware detection system capable of monitoring features (i.e., CPU consumption, number of packets sent over the network, number of running processes, and battery power) and events obtained from the mobile device during execution. Dynamic analysis solves the problems that static analysis faces concerning malware code obfuscation and insufficient detection of dynamic code loading means. Still, it is pretty time-consuming and does not meet the light and fast detection of malware needs. Moreover, our system is running on the user's Android device, and dynamic analysis requires running programs. If dynamic analysis is performed on the user's device, the user's device is already threatened by malware before the analysis is completed, which is unacceptable. Therefore dynamic analysis is not applicable to our approach.

### 2.3. Hybrid Analysis

Many types of malware have the ability to differentiate between environments, which makes dynamics-only analysis much less reliable [21]. Hybrid analytics can be produced by combining static and dynamic analytics. It is a method or technique that integrates run-time data obtained from dynamic analysis with static analysis algorithms used to detect malicious behavior or suspicious functionality, which can compensate for the shortcomings of static and dynamic analysis. Arora et al. [22] proposed a mechanism to detect Android malware from permissions and functions based on network traffic. In this method, permission and network traffic characteristics are used in *FP* growth algorithm to detect malicious behavior. AspectDroid [23] performs static bytecode inspection at the application level and does not require any specific support from the operating system or the Dalvik virtual machine. It monitors the code at compile time using a set of predefined security questions. The target application is then executed on any Android platform of choice and its behavior patterns are dynamically monitored and documented. SamaDroid [24] is a hybrid malware detection model for Android devices. SamaDroid works in two steps. In the first step, static functions are extracted from the source code, such as requested hardware components, requested permissions, used permissions, application components, intention filters and suspicious and restricted API calls, and dynamic functions are collected after execution, such as files generated by the application and network related system calls, such as opening, reading, etc. In the second stage, these static and dynamic features are preprocessed and used as the input of two different machine learning classifiers (such as SVM) to identify whether the application is malware. If the results of static and dynamic analysis are malicious, the application will be regarded as malware. Ahmed et al. [25] proposed a hybrid approach that examined permissions, text and network-based features both statically and dynamically by monitoring memory usage, system call logs and CPU usage, finally, stacked ensemble learning is used to make predictions. Hybrid analysis greatly compensates for the shortcomings of static and dynamic analysis, making the detection effect further improved. However, hybrid analysis also brings a problem that it takes more time than using only static or dynamic analysis. In addition, since dynamic analysis is a part of hybrid analysis, hybrid analysis is not adopted by our method.

### 3. A Lightweight Multi-Source Fast Android Malware Detection Model

To achieve lightweight and fast Android malware detection, we propose a detection method by combining power spectral density, file headers of Dalvik virtual machine executables, and Intent and Permission call features in Android manifest files. Our method is divided into two parts, feature extraction, and classification. The following paper describes the feature extraction scheme, the ensemble model and the base model for each feature in several aspects.

### 3.1. Android Application Structure and Its Feature Selection and Feature Extraction Scheme

As shown in Figure 1, Android application package is a ZIP file with the extension .apk, which contains all the contents of the Android application. Assets stores the static

resource files required by the application, such as images, etc. The resources files in the res directory are compiled into binary to generate the corresponding index IDs in the R.java file. lib directory stores the library files written in C/C++. META-INF stores the signature information of the application, which will be checked before installing the application to ensure the integrity and security of the Android application package. resources.arsc file is used to record the correspondence between the resource files and IDs in the res directory. This model mainly uses the classes.dex and AndroidManifest.xml files.

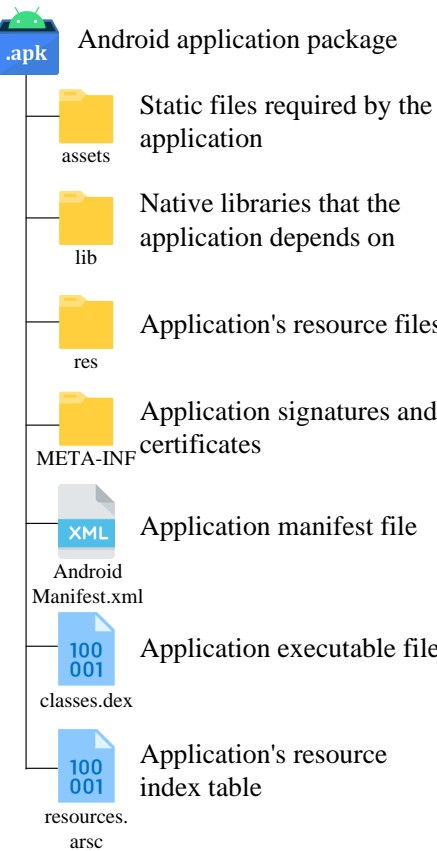

**Figure 1.** The Android application package with the file extension apk is a container file in which all parts of the Android application are packaged.

classes.dex is an executable file for the Android Runtime (ART) and Dalvik virtual machines. The compact Dalvik executable format is designed to work on limited memory and processor speed systems.

classes.dex is essentially all the program logic of an Android application, given that Android applications are typically written in Java and compiled to bytecode by the dx tool. The Java compiler compiles all Java source files to Java bytecode (.class files), and then the dx tool converts them from Java bytecode to Dalvik-compatible bytecode Dex files. The dx tool eliminates the redundant information present in classes, and in dex files, all .class files are wrapped into one file, merging the .class header information and sharing a pool of constants and indexes as in Figure 2. As a result, the vast majority of the code logic of an Android application is contained in the classes.dex file.

Mobile apps frequently request access to sensitive information, such as unique device ID, location data, and contact lists. Android currently requires developers to declare what permissions an app uses [26]. AndroidManifest.xml is the manifest file of the Android application, which describes each component of the application and defines the manifestation of the component such as component name, theme, launch type, operation of the component, launch intent, etc. The manifest file also declares the application properties and permission information.

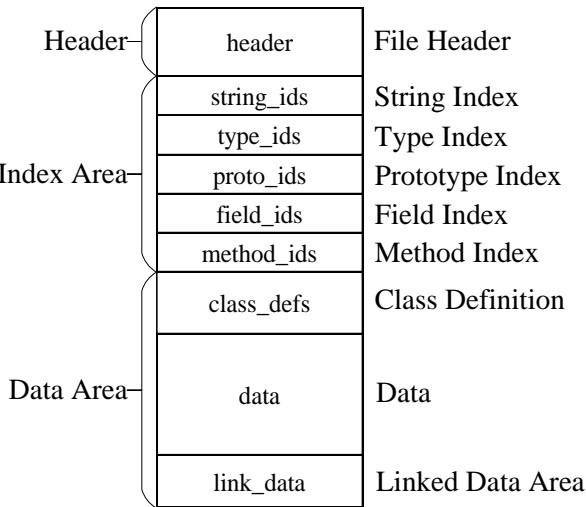

**Figure 2.** The structure of a Dalvik executable.

In summary, the classes.dex and AndroidManifest.xml files contain most of the features of Android applications, so this paper ignores the other files in the APK file and only takes the classes.dex file and AndroidManifest.xml file to extract the features.

### 3.2. Ensemble Model and Base Models

Based on the above, we propose a model to detect Android malware by Dex file header information, power spectrum density information of dex file structure entropy, and permission and intent information in the AndroidManifest.xml file. The structure of the model is shown in Figure 3, this model is divided into four base models and one ensemble learning model.

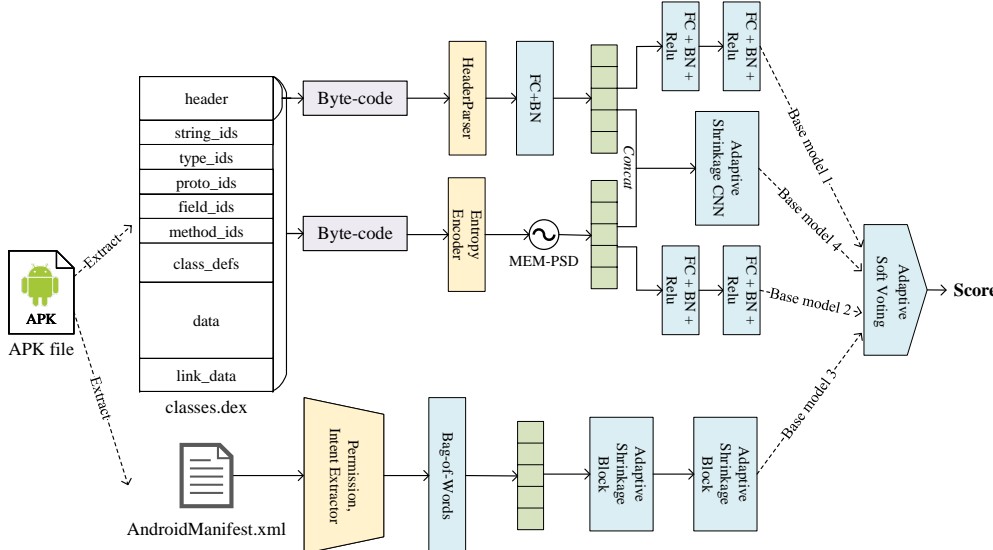

**Figure 3.** The structure of the lightweight multi-source fast Android malware detection model. This model takes the classes.dex and AndroidManifest.xml files in the APK sample as input, followed by feature selection and feature extraction of these two files. The extracted features are predicted by 4 different base models, these base models are integrated by adaptive soft voting method, and finally the probability of the sample being malicious is output.

Specifically, we used the following base models.

- Base model 1 : Abbreviated as $MLP(H)$. It extract and parse the header information of the classes.dex file, encode it as header features, and use the multilayer perceptron for prediction;
- Base model 2: Abbreviated as $MLP(P)$. It calculate the entropy of the classses.dex file, extract the power spectral density of the entropy signal as features by the maximum entropy method, and use the multilayer perceptron for prediction as well;
- Base model 3: Abbreviated as $ASCNN(I)$. perform the prediction on the AndroidManifest.xml file is decoded and parsed. Permission and intent keywords are extracted and encoded as features by bag-of-words model. Since the dimensionality is too high, we use adaptive shrinkage convolutional neural network for dimensionality reduction and then prediction by multilayer perceptron;
- Base model 4: Abbreviated as $ASCNN(C)$. Since theoretically using more base models for ensemble learning will give better results. To improve the ensemble learning, we additionally added a base model that concatenate the features extracted from the DEX file header and the permission and intent features and uses an adaptive shrinkage convolutional neural network and a multilayer perceptron for prediction.

Finally, we use the adaptive soft voting method to perform ensemble learning and predict the final results.

### 3.3. Base Model for Identifying Android Malware by Dex Header

Since the Dex files of different Android applications have different sizes, and some Dex files have large sizes, scanning the entire dex file for all data can be quite time-consuming. A malware detection tool proposed by Zubair et al. for extracting features from PEs completed a single scan of all features in the dataset with a detection rate of over 99% [27]; however, they scanned the entire contents of the PE file, which took nearly an hour. Therefore, we elicit an approach to distinguish malicious and benign applications based only on the header information. Dex headers hold meta-information about a dex file, such as the size and offset information of each index area within the dex, which describes the summary structure of a dex file. In this paper,We analyzed and compared the values of some fields in the dex file header of the dataset malware and benign software samples and presented them by data visualization as shown in Figure 4, we found that malware and benign software have large differences in the distribution of the values of these fields, so we considered that using header information for Android malware classification is effective, while we confirmed this in our experiments.

The above analysis shows significant differences between malware and benign software in the values of several fields in the Dex header. Thus we extract malware features based on the Dex header to discriminate the base model of malware.

The Dex file header parser reads the basic section of the Dex header based on the definition of the dex structure in `dalvik/libdex/DexFile.h` in the Android source code. First, it determines whether it is a valid dex file by checking the Dex magic number field, which the first 8 bytes of the dex file, represented by eight 1-byte unsigned numbers. Its value is a combination of the dex string and the file version number combination, such as "64 65 78 0A 30 33 35 00", using the ASCII table for conversion to get "dex\n\035\0", where the version number is used for the system to identify and parse different versions of the format of The version number is used to provide support for the system to recognize and parse Dex files with different version formats. After judging the legitimate dex file by checking, parse it according to the definition of DexHeader structure in DexFile.h, and get all the fields in the structure. The size and offset information of the parsed file signature, link segment, mapping item, type identifier, string identifier, prototype identifier, etc., are encoded in hexadecimal and normalized to obtain a one-dimensional gray matrix is the feature extracted from the dex header.

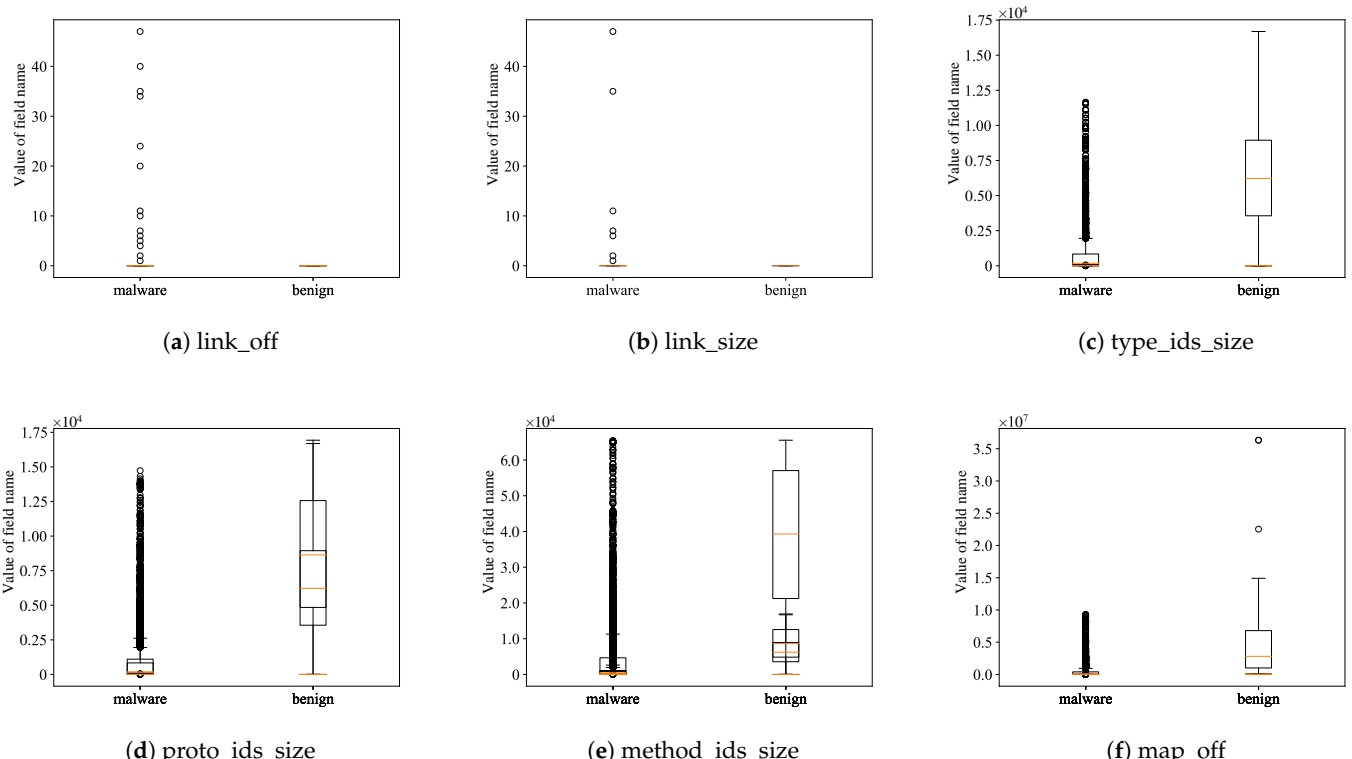

**Figure 4.** Distribution of values in the header of malware and benign software files.

*3.4. Base Model for Identifying Android Malware by Power Spectral Density of Structural Entropy of Dex File*

Entropy measures the randomness or uncertainty of a variable. Shannon relates information to uncertainty; if one can measure the uncertainty of a thing, then one can also measure the amount of information in a given piece of information.

Shannon entropy formula is expressed as Equation (1):

$$H(X) = - \sum_{i=1}^{n} P(x_i) \log P(x_i) \tag{1}$$

where $H(X)$ is the entropy of the variable $X$, $\sum$ defines the sum of the possible values $x_i$ of the variable $X$, and $P(x_i)$ is the probability of occurrence of the possible outcomes $x_i$ of the variable $X$, where $i$ represents the number of outcomes, varying between 1 and $n$.

Malware of the same type usually has similar malicious code segments, which are compiled and eventually reflected in the binary data stream of the Dex file. Therefore, these malicious code fluctuations will also react to the entropy sequence obtained from the binary data stream chunking calculation. The binary stream of the classes.dex file is subjected to the information measure to obtain the entropy sequence. Many scholars have detected the malware based on the entropy features. Wojnowicz et al. [28] proposed a method for the detection of parasitic malware based on the entropy features and achieved good performance. Liu et al. [29] extracted the entropy sequence of malicious documents and detected the malware based on the machine learning algorithms.

Unlike previous studies, we use structural entropy for the detection against classes.dex files, and also because the size of the classes.dex file of different Android programs is inconsistent. The length of the received entropy sequence is also varying. So we use the power spectral density method to calculate the power spectral density of the entropy sequence to quickly obtain the fixed-length features for machine learning.

The power spectral density of a signal (Power Spectral Density) describes the power present in a signal as a function of frequency per unit. The calculation methods of power spectrum mainly include the fast Fourier transform method, Welch method, multi-window method, maximum entropy method, etc. The first three belong to the periodogram method. Since the periodogram method is a method to estimate the finite length autocorrelation of the signal, it requires truncation or windowing of the signal sequence so that the estimated power spectrum is the convolution of the true spectrum of the signal sequence and the window spectrum, so its ability to produce accurate power spectrum estimates is limited [30], which is why we use the maximum entropy method to calculate the power spectral density $P(f)$ [31], which is expressed as Equation (2):

$$P(f) = \frac{P_m \triangle t}{|1 + \sum_{k=0}^{m} \gamma_{m,k} exp(-i2\pi f k \triangle t)|^2} \tag{2}$$

where $P(m)$ is the output power of the filter at the fluctuation period of the $m$th order, $\gamma_{m,k}$ is the filter coefficient when $m = 0, 1, \ldots, M$, and $M$ is the corresponding filter coefficient at the optimal filter order, and $P_m$ and $\gamma_{m,k}$ are obtained by solving the Yule-Walker equation through Burg method [32,33].

We randomly selected a benign sample and a malware sample and calculated the structural entropy sequences of the binary streams of their Dex files and power spectral densities of the entropy sequences, which we abbreviate as MEM-PSD, as shown in Figure 5, from which we can learn that different lengths of structural entropy sequences can be calculated for a fixed-length power spectral density.

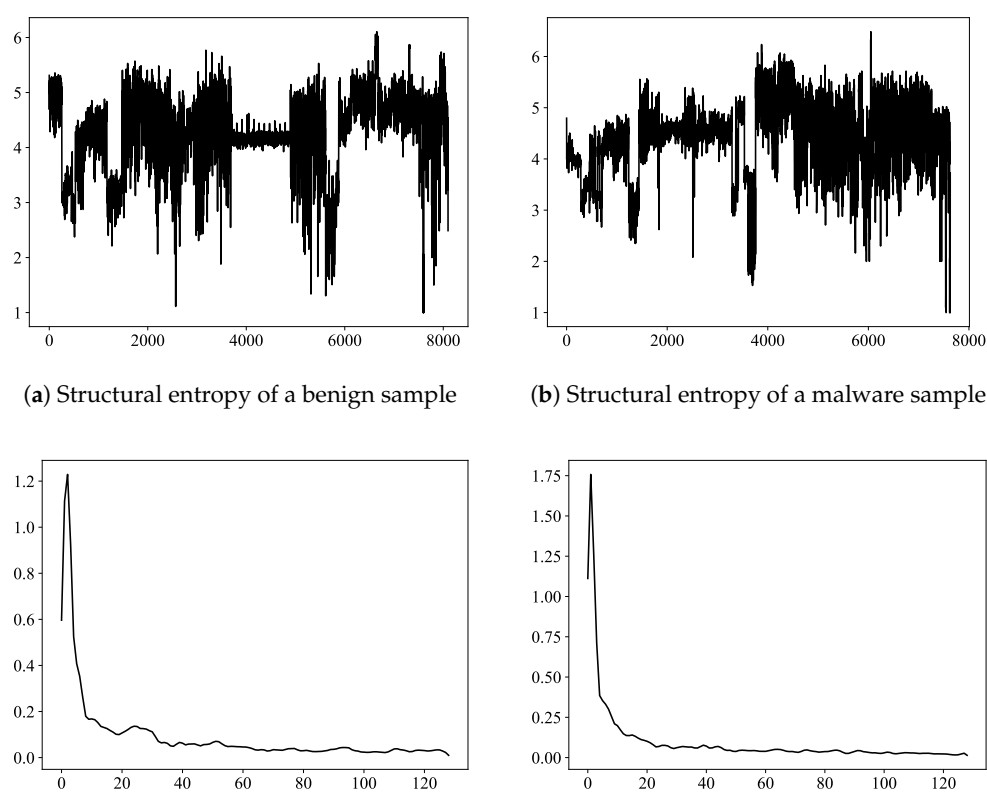

(**a**) Structural entropy of a benign sample

(**b**) Structural entropy of a malware sample

(**c**) MEM-PSD of a benign sample

(**d**) MEM-PSD of a malware sample

**Figure 5.** Distribution of structural entropy and MEM-PSD of benign software samples and malware samples.

We take the byte stream of classes.dex as a time series, divide the sequence with a block size of 256 bytes, and then construct a structural entropy sequence according to

Algorithm 1. We compute the power spectral density of the structural entropy sequence to obtain a power spectral sequence of length 128 as a feature vector. This feature vector is used as input and the multilayer perceptron is used as a classifier to construct this base model.

---

**Algorithm 1:** Calculate the structural entropy sequences using Shannon entropy

---

　　**Input:** byte sequence $S_b$, block size $bs$
　　**Output:** structural entropy sequence $S_H$
1　$L_{sb}$ = Length of $S_b$;
2　$Left = L_{sb} \bmod bs$;
3　**if** $Left < bs/2$ **then**
4　　Truncate the end of $S_b$ by the length $Left$;
5　**else if** $Left > bs/2$ **then**
6　　Pad the end of $S_b$ with an all-zero sequence of length $bs - Left$;
7　**end**
8　$X$ = Reshape $S_b$ to a 2D matrix of size $(L_{sb}|256 + 1256)$;
9　Initialize a one-dimensional array $S_H$;
10　**for** *Row in first dimension of X* **do**
11　　Calculate the number of occurrences of each value in *Row* and assign the result to $C$;
12　　$P = C/bs$;
13　　Filter out the elements of $P$ equal to 0;
14　　$H = ShannonEntropy(P)$;
15　　$S_H$=Append $H$ to $S_H$;
16　**end**

---

### 3.5. Base Model for Identifying Android Malware by Permission and Intent

Permission control is a key problem in the security of the Android operating system. Android permissions enforce the restrictions on the specific operation to offer concrete security features [34].

The AndroidManifest.xml holds information about the application structure and is organized in the form of Components. Android Framework defines four kinds of Components, namely Activity, Service, Broadcast Receiver, and Content Provider. The manifest file also contains the list of permissions which are requested by the application to work and needed to access its components [35].

Usually, AndroidManifest.xml is encrypted, we extract the application permissions usage, components, and intent by decrypting the file into a legal XML document and parsing it. The extracted information is used to construct a feature vector.

As shown in Figure 6, malicious applications make intensive use of some specific permissions. They are more homogeneous in their functionality than general applications, requiring only a combination of specific permissions. The analysis shows that this information in the manifest file plays a crucial role in determining the type of application.

Since the set of permission and intent keywords is small, we use a sparse expression to reduce the model complexity, using the idea of the Bag of word model, which treats the occurrence of each permission and intent keyword as an independent probability. All permissions and intentions of the dataset are extracted and filtered to sieve out keywords with frequencies less than 2, constituting a lexicon containing $N$ keywords. The keywords are removed from the list file of each application and encoded using Bag of Word model. UNK replaces the keywords not in the dictionary to form an $N + 1$-dimensional feature vector. Since the dimensionality of the bag-of-words vector is equal to that of the dictionary (the number of words in the dictionary), the bag-of-words vector is also sparse, with often only a few tens or hundreds of non-zero items in thousands or even tens of thousands of dimensions. In our model, we designed a dictionary of size 4380. If we directly use a

multilayer perceptron to predict this feature, we will generate a large number of redundant parameters, which will greatly increase the size of our model, so we use a convolutional network to further extract and optimize the feature, and at the same time reduce its dimensionality so that the classifier can classify it quickly. We constructed a shallow convolutional network using three layers of adaptive shrinkage convolution as shown in Figure 7. This network finally outputs 128-dimensional feature vectors, which are finally classified by a multilayer perceptron. Next, we explain the adaptive systolic convolution unit in detail in the next subsection.

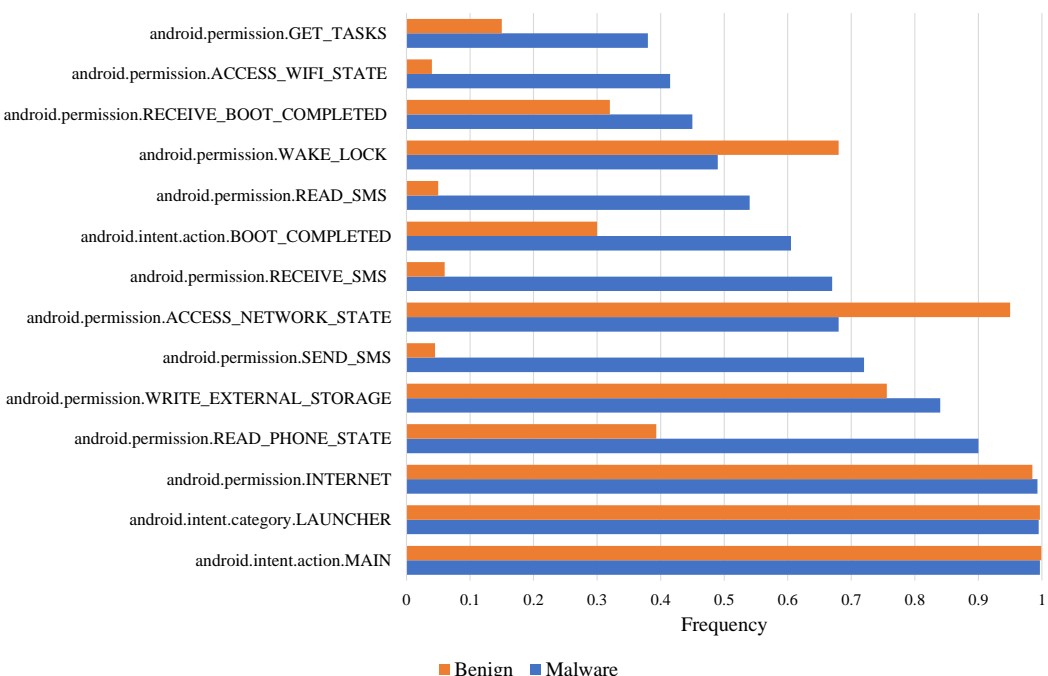

**Figure 6.** Statistics of permissions and intent used by benign software samples versus malware samples.

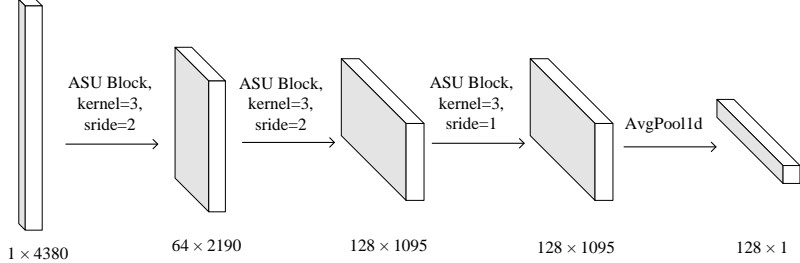

**Figure 7.** Convolutional network structure and feature dimension.

### 3.6. Adaptive Shrinkage Convolution Unit

For many Android malware applications, malicious code fragments are mixed into a large number of normal code fragments. As a result, many noises unrelated to malicious code fragments appear in the extracted features. The features need to be noise-reduced to improve the feature learning ability to address this problem. The classical wavelet threshold noise reduction method consists of three main steps: wavelet decomposition, soft thresholding, and wavelet reconstruction; in this noise reduction method, it is a challenging problem to construct a suitable filter operator to set a reasonable soft threshold. An adaptive shrinkage convolution unit is proposed in this paper to solve this problem, its specific structure is shown in Figure 8.

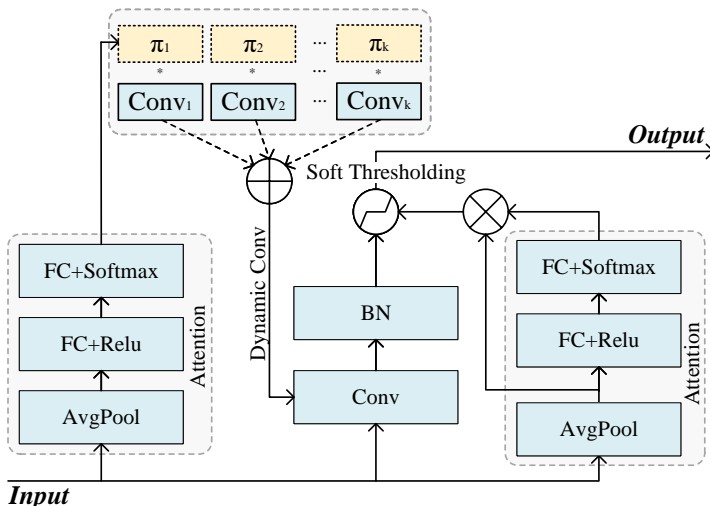

**Figure 8.** Adaptive shrinkage convolution unit structure.

The convolutional layer is used to compute the output feature mapping by convolving the feature mapping of the previous layer with a set of filters. These filters are the only parameters of the convolutional layer and are usually learned in training by a backpropagation algorithm. This model makes two main improvements on the traditional convolutional layer, which uses convolutional kernels (filters) learned in training and kept constant in testing. In contrast, our model uses convolutional kernels that change during testing as the input varies. This is achieved by learning the kernel functions that map the inputs to the convolutional kernels through an attention mechanism. Meanwhile, the key of feature learning method is not only to extract the target information related to the labels, but it is also important to eliminate irrelevant information, so it is important to introduce soft thresholding inside the deep neural network to adaptively eliminate redundant information during the feature learning process and improve the learning of useful features. More importantly, each sample should have a different threshold value. This is because a sample set often contains many samples, and the amount of noise contained in these samples is often different. In deep learning algorithms, the size of the threshold cannot be interpreted because these features have no clear physical meaning, but the reasoning is similar, each sample should have a different threshold, so this model also uses the attention mechanism to learn to map the input to the threshold of the soft-threshold activation function.

The model representation for each convolutional layer is enhanced by superimposing convolutional kernels nonlinearly according to attention by the dynamic convolution [36,37] method. Let the traditional static convolution be represented as $y = g(W^T(x)x + b)$, where $W$ and $b$ are the weight matrix and bias vector, and $g$ is an activation function, and define the dynamic convolution by aggregating multiple ($K$) linear functions $\tilde{W}_k^T x + \tilde{b}_k$ as in Equations (3)–(5):

$$y = g(\tilde{W}^T(x)x + \tilde{b}(x)) \tag{3}$$

$$\tilde{W}(x) = \sum_{k=1}^{K} \pi_k(x)\tilde{W}_k \tag{4}$$

$$\tilde{b}(x) = \sum_{k=1}^{K} \pi_k(x)\tilde{b}_k \tag{5}$$

where $\pi_k$ is the attention weight of the $k$th linear equation $\tilde{W}_k^T x + \tilde{b}_k$, $\pi_k(x) \in [0,1]$ and $\sum_{k=1}^{K} \pi_k(x) = 1$, the weight $\tilde{W}(x)$ and the bias $\tilde{b}(x)$ are functions of the inputs and share the same attention. The attention weights $\pi_k(x)$ are not fixed but vary for each input $x$, and they represent the best aggregation of a linear model $\tilde{W}_k^T + \tilde{b}_k$ for a given input.

The aggregated model $\tilde{W}^T(x)x + \tilde{b}(x)$ is nonlinear. Therefore, it is possible to change the convolution kernel weights adaptively according to the input $x$ to make it more expressive compared to static convolution.

Meanwhile, in this paper, based on the study of deep residual shrinkage networks [38], a soft thresholding activation function is introduced to set the features corresponding to these noises to zero with the help of soft thresholding and dynamically set the thresholds for each sample individually according to each sample through an attention mechanism, integrated into this unit as a nonlinear transformation layer. Let $x$ be the input feature, $y$ be the output feature, and $\tau$ be the threshold value. The formula for soft thresholding is expressed as Equation (6).

$$y = \begin{cases} x - \tau & x > \tau \\ 0 & -\tau \leq x \leq \tau \\ x + \tau & x < -\tau \end{cases} \tag{6}$$

This unit improves the expressiveness of the convolutional network, at the same time, it seeks a balance between the performance of the network and the computational load. The computational complexity of the convolutional kernel and activation function increases slightly, but the kernel function and activation function of the convolutional kernel need to be computed only once, and the computational resources used are much smaller than those of using more layers of convolution, thus reducing the complexity of the overall model.

*3.7. Adaptive Soft Voting Ensemble Method*

An ensemble learning method combines the results of two or more separate machine learning algorithms and attempts to produce results that are more accurate than any single algorithm.

Voting is a combination strategy for classification problems within ensemble learning. The basic idea is to select the class with the most output among all machine learning algorithms. There are two types of machine learning algorithm outputs for classification: one is the direct output of class labels, and the other is the output of class probabilities, using the former for voting is called Hard voting, and using the latter for classification is called Soft voting.

Soft voting obtains the weighted average of each class probability by inputting weights, and selects the class with the more significant value; soft voting returns the class labels as Argmax of the sum of predicted probabilities, which is achieved by outputting class probabilities. We propose the soft voting method using the above base models for ensemble learning. The weights of traditional static weighted average probability soft voting are determined manually. Generally, they use multiple homogeneous base models, but we extract different features using different models. Since different features and models have different training curves and fitting abilities, the voting weights need to be dynamically adjusted during the training process. So we propose a dynamic weighted soft voting method, where the gradient descent principle automatically determines the voting weights, and a new loss function is designed for this model.

Let $N$ be the number of samples, $p_i$ be the probability that the $i$th sample is a positive case, and $y_i$ be the label of the $i$th sample, then the binary cross-entropy formula is as Equation (7).

$$L_{BCE}(p, y) = \frac{1}{N} \sum_{i=1}^{N} -[y_i \cdot log(p_i) + (1 - y_i) \cdot log(i - p_i)] \tag{7}$$

where the probabilities $[p_1, p_2, \cdots, p_j]$ of the positive cases derived from $M$ models and the weights $W$ are calculated by the Hadamard product of Softmax operations to derive the probability of positive cases $p_{vote}$ after soft voting, and the Softmax function $\sigma(W)$ and $p_{vote}$ is calculated as Equations (8) and (9).

$$\sigma(W)_j = \frac{e^{W_j}}{\sum_{m=1}^{M} e^{W_m}} \qquad j = 1, 2, \cdots, M \tag{8}$$

$$p_{vote} = \sum_{j=1}^{M} ([p_1, p_2, \cdots, p_j] \circ \sigma(W)) \tag{9}$$

The binary cross-entropy loss $[l_1, l_2, \cdots, l_j]$ derived from $M$ models for Softmax operation and the absolute value of the difference between the value of the weights after Softmax operation are calculated as the average and multiplied by the value obtained from the weight parameter $\mu$. Finally, the sum of this value and the binary cross-entropy of the soft voting results is calculated as the loss *Loss* of adaptive soft voting, which is calculated as Equation (10).

$$Loss = L_{BCE}(p_{vote}, y) + \mu \times \frac{\sum_{j=1}^{M} |\sigma(-[l_1, l_2, \cdots, l_j]) - \sigma(W)|}{M} \tag{10}$$

This loss function can effectively suppress the problem that the adaptive soft voting method assigns too small weights to the base model with slow gradient decline during the training process so that the output of the base model is not considered even if the base model achieves good results at the later stage of training, leading to the problem of falling into a local optimum.

## 4. Experimental Results and Analysis

In this section, we first introduce the dataset, experimental environment for the experiments. Then, we compare our model with other benchmark models through several experiments to examine the performance of each module of this model, and finally we analyze the experimental results and give the experimental conclusions.

### 4.1. Dataset

Three publicly available datasets were used in this experiment, the CICMalDroid2020 dataset [39] and CIC-InvesAndMal2019 dataset [40] from the Canadian Institute for Cybersecurity Research (CIC) and the Drebin (2012) dataset [10] from the Institute for Systems Security at the Technical University of Braunschweig: the CICMalDroid 2020 dataset has more than 17,341 Android samples, including VirusTotal Service, Contagio Security Blog, AMD, Maldozer and other datasets used in recent research contributions; CIC-InvesAndMal2019 contains 5491 samples (426 malware and 5065 benign software); the Drebin dataset contains 5560 malware samples. Since the sample size of individual datasets is too small, we decided to combine them to build a larger dataset.

### 4.2. Experimental Setup

This experiment is trained on a Tesla V100 GPU (16 GB) using the Pytorch 1.9.0 framework in Centos7 and Cuda10.2 environments. The experiment sets the batch size to 16, the momentum of stochastic gradient descent SGD to 0.9, the learning rate to $0.5 \times 10^{-2}$, and the multiplicative factor of learning rate decay to 0.5.

Three metrics, area under the ROC curve (AUC), which formula is Equations (11) and (12), accuracy (ACC), which is expressed as Equation (13), and the summed mean of precision and recall (F1-Score), which is expressed as Equation (14), are selected to evaluate the model performance.

$$AUC = \frac{\sum I(P_{pos}, P_{neg})}{M \cdot N} \tag{11}$$

$$I(P_{pos}, P_{neg}) = \begin{cases} 1 & P_{pos} > P_{neg} \\ 0.5 & P_{pos} = P_{neg} \\ 0 & P_{pos} < P_{neg} \end{cases} \tag{12}$$

where $M$ is the number of positive samples (malware), $N$ is the number of negative samples (benign software), so there are $M \cdot N$ pairs of samples in the data set. $P_{pos}$ is the prediction probability of positive samples, and $P_{neg}$ is the prediction probability of negative samples.

$$\text{ACC} = \frac{TP + TN}{TP + TN + FP + FN} \tag{13}$$

$$\text{F1} = \frac{N - TN}{N + TP - TN} \tag{14}$$

where $TP$ is the number of results that correctly predicted that the sample is malware, $TN$ is the number of results that correctly predicted that the sample is not malware, $FP$ is the number of results that incorrectly predicted that the sample is malware, and $FN$ is the number of results that incorrectly predicted that the sample is not malware.

### 4.3. Comparison of Different Methods

The methodology in this paper is compared with several recent benchmark models for detecting Android malware, and their brief descriptions and experimental results are given below.

- Meenu's method: CNN-Based Android Malware Detection [41]. It Extracts permission information from AndroidManifest.xml, encodes it into a permission vector, and extracts features using LeNet.
- XushengXiao's method: An Image-Inspired and CNN-Based Android Malware Detection Approach [42]. It reads Dalvik bytecode in hexadecimal, transforms it into a three-channel color matrix, and extracts features using CNN.
- Muhammad's method: Static Malware Detection and Attribution in Android Bytecode through an End-to-End Deep System [43]. It proposes an end-to-end network to detect the byte-code of an application by using a bidirectional LSTM on the extracted opcodes to detect Android malware by using bi-directional LSTM.
- David's method: EntropLyzer: Android Malware Classification and Characterization Using Entropy Analysis of Dynamic Characteristics [44]. It proposes an entropy-based behavior analysis technique using memory, API, network, Logcat, and battery dynamic characteristics to classify and characterize Android malware.
- XushengWang's method: MFDroid: A Stacking Ensemble Learning Framework for Android Malware Detection [13]. It uses seven feature selection algorithms to select permissions, API calls and opcodes, then merges the results of each feature selection algorithm to obtain a new feature set, and subsequently uses logistic regression to obtain classification results.
- Mahindru's method: HybriDroid: an empirical analysis on efective malware detection model developed using ensemble methods [12]. It applies five distinct machine learning algorithms and non-linear ensemble decision tree forest to detect malware in Android applications.
- Ahmed's method: Mitigating adversarial evasion attacks of ransomware using ensemble learning [25]. It proposes an hybrid analysis approach to detect Android malware by monitoring memory usage, system call logs and CPU usage, statically and dynamically checking permissions, text and network-based functions.
- Ruitao's method: A Performance-Sensitive Malware Detection System Using Deep Learning on Mobile Devices [5]. It proposes a fast malware detection method by extracting manifest properties and API calls directly from the binary code of an Android application and vectorizing them, and finally using a quantized neural network.

The performance of the model is shown in Table 1, and the best results are bolded in the table. Considering the metrics of AUC, ACC, F1-Score, and average time consumption, our method achieves a better trade-off in accuracy and speed than several other methods. The time consumption of methods using dynamic or hybrid analysis is not included in the table.

**Table 1.** Comparison of test results of different methods on PC.

| Method | AUC | ACC | F1-Score | Average Time Consumption on PC |
|---|---|---|---|---|
| Meenu's [41] | 95.36% | 93.67% | 94.53% | 0.64 s |
| XushengXiao's [42] | 94.34% | 93.00% | 94.02% | 0.22 s |
| XushengWang's [13] | 97.66% | 96.35% | 96.10% | 0.21 s |
| Muhammad's [43] | 98.82% | **99.92%** | **98.35%** | 0.35 s |
| David's [44] | 99.06% | 98.42% | 98.20% | - |
| Mahindru's [12] | 98.95% | 98.53% | 96.72% | 0.15 s |
| Ahmed's [25] | 99.38% | 98.77% | 98.12% | - |
| Ruitao's [5] | 97.06% | 96.75% | 96.91% | 0.19 s |
| MSFDroid | **99.52%** | 97.26% | 97.89% | **0.14 s** |

One of them, Ruitao's method, has similar goals as our method to build lightweight detection methods that can run with Android devices, so we delved into the gap in time efficiency between our method and Ruitao's. We tested on different devices while recording the average detection time of the detected samples, and the statistics are shown in Table 2. Our method achieves more accurate detection results with faster detection speed on multiple test platforms compared to Ruitao's.

**Table 2.** Comparison of our method with Ruitao's in terms of time efficiency.

| Device | CPU | | Time Consumption | |
|---|---|---|---|---|
| | Model | Performance | MSFDroid | Ruitao's [5] |
| Galaxy J7 Pro | Exynos 7870 Octa | 448 | 2.33 s | 3.96 s |
| Nexux 6P | Qualcomm Snapdragon 810 | 514 | 1.57 s | 2.20 s |
| Oppo F3 | Mediatek MT6750 | 668 | 1.36 s | 1.76 s |
| OnePlus 3 | Qualcomm Snapdragon 820 | 759 | 1.05 s | 1.65 s |
| OnePlus 5T | Qualcomm Snapdragon 835 | 1627 | 0.81 s | 1.03 s |
| Huawei P30 | HiSilicon Kirin 980 | 2419 | 0.37 s | 0.46 s |
| OnePlus 8Pro | Qualcomm Snapdragon 865 | 3045 | 0.28 s | 0.38 s |

The total time consumption includes the extraction and prediction times. The extraction time is the time consumption for decompressing the APK, calculating the structural entropy of the Dalvik binary, calculating the maximum entropy-power spectral density, decoding the Android manifest file, and calculating the Bag of word model. The prediction time includes the calculation time of the base model and the calculation time of the soft voting ensemble model as described above. We selected seven devices for testing, including five real Android devices based on ARMv8 architecture with different generations of releases, and used two x86_64 architecture computers for comparison.

Our research found that time consumption is mainly related to CPU performance. It is difficult to measure CPU performance by design parameters due to inconsistencies in the CPU process, architecture design, base clock speed, and turbo boost clock speed. Therefore, we used GeekBench5 to measure CPU performance and plotted Figure 9, which includes the CPUs used in our method and Ruitao's. It reflects the single-core and multi-core performance differences of different CPUs. The results of Table 3 and Figure 9 together reflect that the multi-core performance of CPUs mainly influences the time consumption of our method.

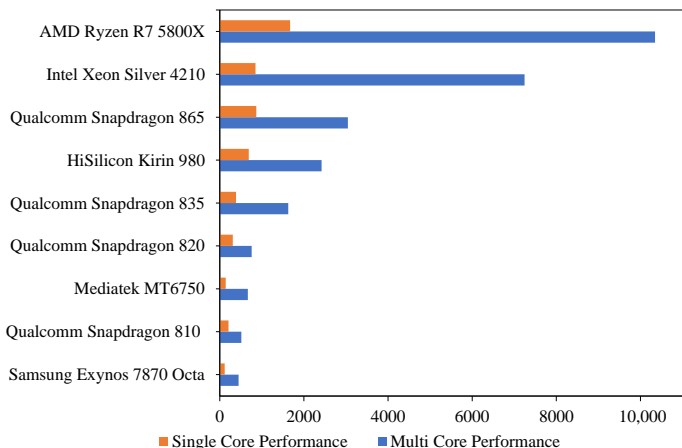

**Figure 9.** CPU benchmarks

**Table 3.** Efficiency of running on different devices.

| Device | CPU | Arch | Extraction Time | Prediction Time | Total Time |
|---|---|---|---|---|---|
| Galaxy J7 Pro | Exynos 7870 Octa | ARMv8 | 476.85 s | 1758.161 s | 2.232 s |
| Nexus 6P | QCOM Snapdragon 710 | ARMv8 | 392.412 s | 1183.588 ms | 1.575 s |
| Oppo F3 | Mediatek MT6750 | ARMv8 | 375.939 ms | 987.057 ms | 1.363 s |
| OnePlus 3 | QCOM Snapdragon 820 | ARMv8 | 257.021 ms | 793.548 ms | 1.051 s |
| OnePlus 5T | QCOM Snapdragon 835 | ARMv8 | 211.628 ms | 699.563 ms | 0.911 s |
| Huawei P30 | HiSilicon Kirin 980 | ARMv8 | 176.942 ms | 195.222 ms | 0.372 s |
| OnePlus 8Pro | QCOM Snapdragon 865 | ARMv8 | 96.501 ms | 182.123 ms | 0.279 s |
| x64 Server | Intel Xeon Silver 4210 | x86_64 | 67.515 ms | 89.088 ms | 0.157 s |
| x64 PC | AMD Ryzen7 5800X | x86_64 | 58.019 ms | 46.429 ms | 0.104 s |

Table 2 and Figure 10 shows the time consumption of our method and Ruitao's on devices with different CPU performances. CPU performance is referenced to GeekBench5's multi-core score results, time consumption is calculated in seconds, and the data are fitted using a power function. To reduce the impact on Android malware detection, we repeat the detection five times for each APK file. Figure 10 reflects that the predicted time consumption of our method is generally lower than Ruitao's on different devices.

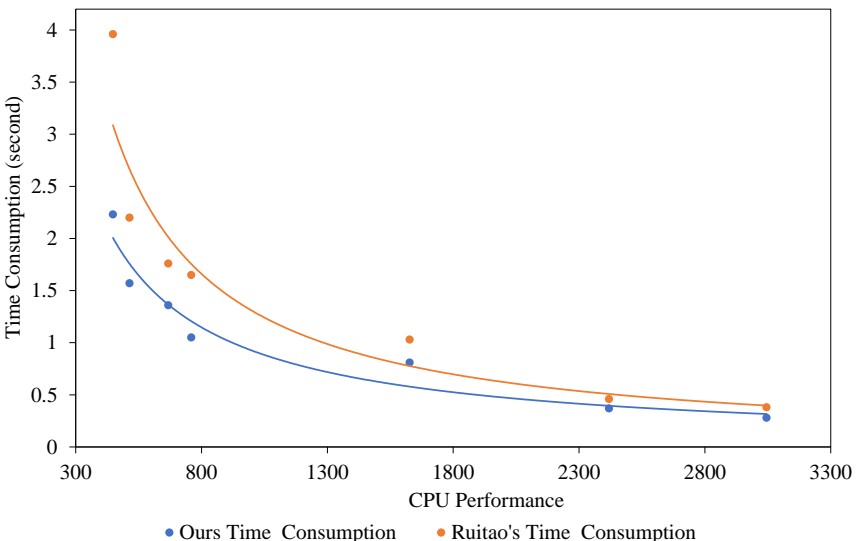

**Figure 10.** The relationship between processor performance and time consumption.

Meanwhile, we compare the number of model size and accuracy of our method and Ruitao's as shown in Figure 11. We use different base models for ensemble learning. Under the condition of achieving the same or better performance, the number of parameters of our model is much smaller than the number of parameters of several models given by Ruitao's and even better than the quantized results of Ruitao's.

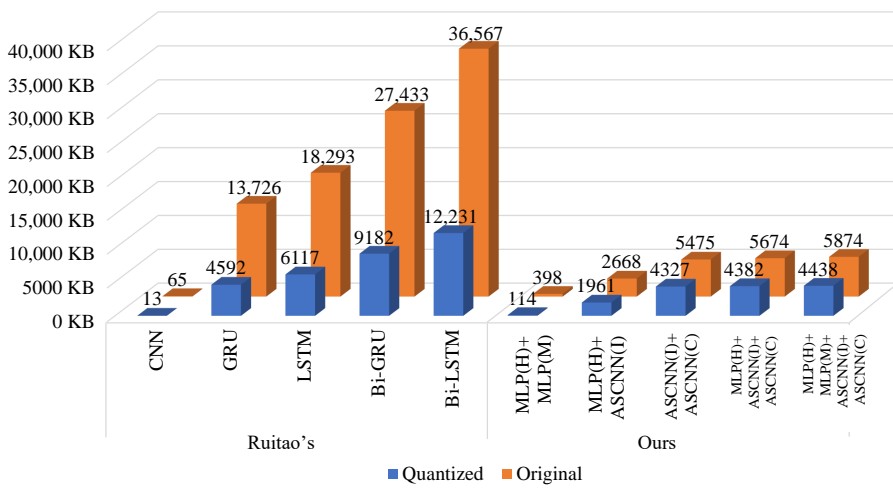

**Figure 11.** Comparison of model size.

Since we use multiple base models and different features, to reduce the complexity of the representation, we define the abbreviations of models and features as follows. *AdaSV* is adaptive soft voting, *ASCNN* is adaptive shrinkage convolutional neural network, and *MLP* is multilayer perceptron. *H* is Dex head feature, *M* is power spectral density feature of entropy sequence of Dex file, *I* is permission and intent feature, and *C* is the combined features of Dex head features and power spectral density features of entropy sequences of Dex files.

In summary, our method has a more significant advantage in terms of performance, time efficiency, and space efficiency by comparing several dimensions.

### 4.4. Comparison with Anti-Virus Softwares

Compared with other methods in the literature, our method shows competitiveness in terms of performance and efficiency. However, it should eventually be compared with anti-virus products commonly used in the industry to show its practical value.

We selected five free antivirus software on the market and used the 5217 malware samples for testing. The experimental results are shown in Table 4.

**Table 4.** Comparison with anti-virus softwares.

| Anti-Virus Software | Detection Rate | Average Time Consumption per Sample |
| --- | --- | --- |
| Huorong | 86.4% | 0.07 s |
| Norton | 93.7% | 0.06 s |
| McAfee | 93.8% | 0.05 s |
| Kaspersky | 94.3% | 0.07 s |
| Avast | 96.7% | 0.05 s |
| MSFDroid | 98.6% | 0.14 s |

In this study, we observed a large difference in the detection effectiveness of these five anti-virus software, which we believe is mainly due to their different virus libraries, which are not specifically designed to detect Android malware. The detection rate of Avast, which is the best detection, is 96.7%, and our method MSFDroid achieves a detection rate of 98.6%.

This proves that our method is more effective with the fingerprint database used by many anti-virus software.

### 4.5. Analysis of Experimental Results

This section compares and analyzes the performance of different base models, adaptive shrinkage convolution neural network, and adaptive soft voting.

#### 4.5.1. A Study on the Performance of Adaptive Shrinkage Convolution

The adaptive shrinkage convolutional neural network improves performance compared to the conventional convolutional network using the same number of convolutional layers. This result in Table 5 shows that the attention mechanism is used to dynamically adjust the convolutional kernel's weights and the activation function's threshold according to the noise level. Although the introduced attention mechanism increases the size and complexity of the convolutional kernel generating function and activation function and increases the computational effort, the additional computational step can be neglected compared to the convolutional computation because the convolutional kernel parameters and the activation function threshold are only computed once. Therefore, the traditional convolutional neural network achieves higher performance with fewer convolutional layers and reduces the overall network computation.

**Table 5.** Performance comparison of adaptive shrinkage convolution neural network and conventional convolutional network.

| Model | AUC | ACC | F1-Score |
|---|---|---|---|
| 3-layer convolutional neural network | 97.26% | 92.57% | 94.76% |
| 6-layer convolutional neural network | 98.74% | 95.15% | 95.61% |
| 3-layer adaptive shrinkage convolution neural network | 99.23% | 95.28% | 96.29% |

#### 4.5.2. A Study on the Performance of Adaptive Soft Voting Method

As shown in Table 6, by comparing the performance of different base models and adaptive soft voting assembled with multiple base models, a single base model has a limited performance on malware detection. Still, as shown in Table 7, by assembling with the adaptive soft voting method, we achieve a maximum improvement of +5%, +13%, +16% by reaching 99.52%, 96.97%, 97.89% in three performance evaluation metrics. The adaptive soft voting method assembles multiple base models. It achieves significant performance improvements, while the performance of adaptive soft voting improves more with the increase of the number of integrated base models.

**Table 6.** Performance between different base models and ensemble learning models.

| Integration of Base Models | AUC | ACC | F1-Score |
|---|---|---|---|
| MLP(H) | 95.73% | 83.54% | 81.07% |
| MLP(M) | 96.12% | 88.98% | 91.76% |
| ASCNN(I) | 98.06% | 88.97% | 88.75% |
| MLP(H) + MLP(M) | 97.66% | 94.30% | 95.94% |
| ASCNN(I) + ASCNN(C) | 99.23% | 95.69% | 96.76% |
| MLP(H) + MLP(M) + ASCNN(I) | 99.39% | 96.83% | 97.20% |
| MLP(H) + MLP(M) + ASCNN(I) + ASCNN(C) | 99.52% | 97.26% | 97.89% |

We use different ensembles of base models to compare the performance of adaptive soft voting with static weighted soft voting, respectively. Under the condition of the dynamic weighted soft voting weight parameter $\mu = 0.5$, the ensembles using different base models improves in three performance metrics, including a maximum improvement of 3.49% in ACC, 2.58% in F1-Score, and 0.43% in AUC.

**Table 7.** Comparison of adaptive soft voting and static weighted soft voting.

| Integration of Base Models | Ensemble Model | AUC | ACC | F1-Score |
|---|---|---|---|---|
| MLP(H)+MLP(M) | Adaptive Soft Voting ($\mu = 0.6$) | 97.66% | 94.30% | 95.94% |
| | Static Weighted Soft Voting | 97.14% | 93.12% | 95.13% |
| ASCNN(I)+MLP(M) | Adaptive Soft Voting ($\mu = 0.6$) | 99.23% | 95.28% | 96.29% |
| | Static Weighted Soft Voting | 99.13% | 93.86% | 95.37% |
| MLP(H)+MLP(M)+MLP(I) | Adaptive Soft Voting ($\mu = 0.6$) | 99.39% | 96.83% | 97.20% |
| | Static Weighted Soft Voting | 99.15% | 94.06% | 95.79% |
| MLP(H)+MLP(M)+ASCNN(I)+ ASCNN(C) | Adaptive Soft Voting ($\mu = 0.5$) | 99.52% | 97.26% | 97.89% |
| | Static Weighted Soft Voting | 99.09% | 93.49% | 95.42% |

To investigate the reason for the performance difference between adaptive soft voting and soft voting methods with static weights, we use the ensemble of two base models, MLP(I), a multilayer perceptron with Intent & Permission features as input, and MLP(M), a multilayer perceptron with MEM-PSD features as input, and their AUC, Loss, and soft voting weights during the training process is shown in the Figures 12 and 13. It can be seen from the figures that MLP(I) has more jitter during the training process, MLP(M) has less jitter but the final performance is lower than MLP(I). In the traditional soft voting method, which cannot adjust the weight of each base model, we can see from the training curve in the Figure 12, the model with more jitter even becomes noise, which leads to a negative impact on the decision making of the voting algorithm, making its AUC curve less flat and underperforming than adaptive soft voting during the training process.

Our proposed adaptive soft voting method continuously adjusts the weights according to the performance of each base model during the training process, effectively avoiding the jitter problem caused by some base models and the noise generated by the poor performance of some base models, making the adaptive soft voting method effectively adapt to different base models and significantly improving the performance of soft voting.

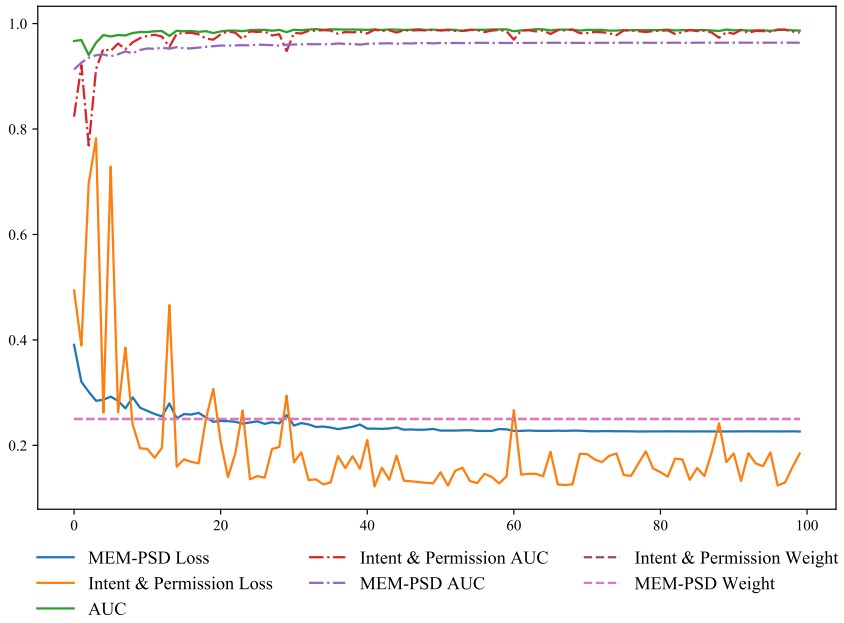

**Figure 12.** Static weighted Soft Voting.

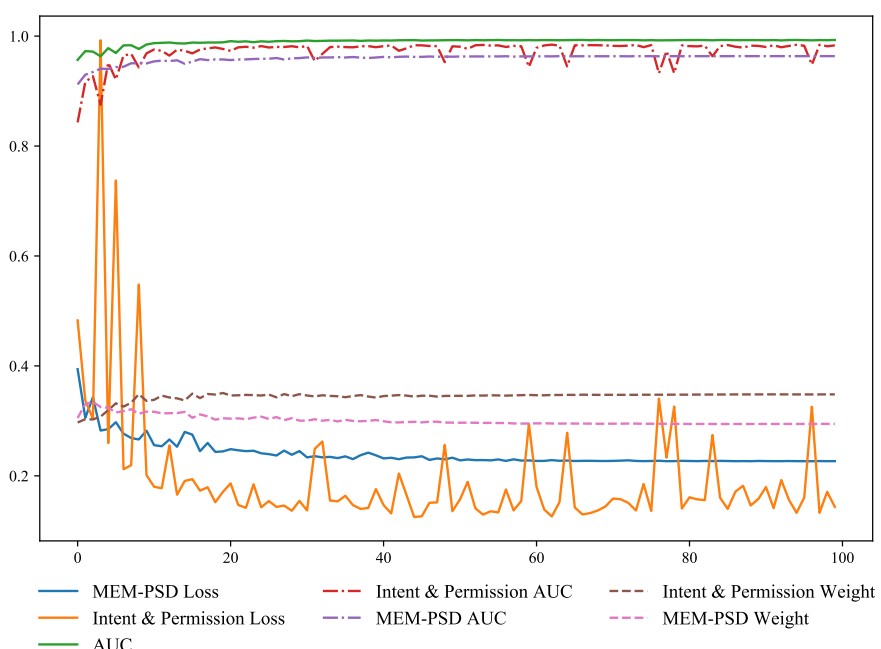

**Figure 13.** Adaptive Soft Voting ($\mu = 0.6$).

### 4.5.3. A Study of Weighting Parameter in Adaptive Soft Voting Loss Functions

The model's accuracy is obtained by adjusting the weight parameter $\mu$ of adaptive soft voting, conducting several experiments, and plotting the scatter plot as in Figure 14. As $\mu$ increases continuously, the accuracy peaks around $\mu = 0.8$ and decreases. When $\mu = 0$, the loss function cannot penalise the weights with large values, thus making the weights almost completely concentrated in the base model with the best effect, as shown in Figure 15. This makes the soft voting fall into a local optimum and makes the overall performance of integrated learning drop significantly.

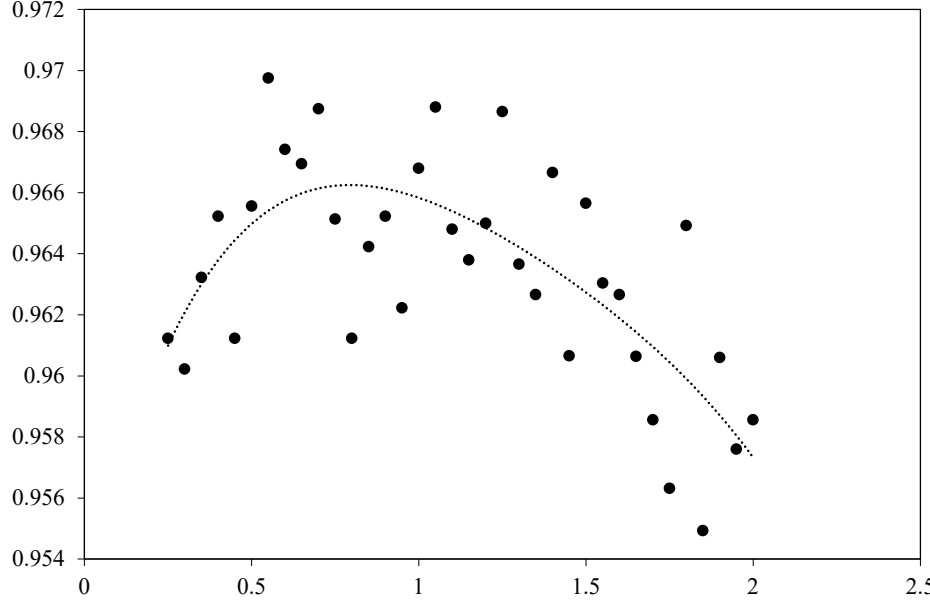

**Figure 14.** Relationship between adaptive soft voting weights and performance.

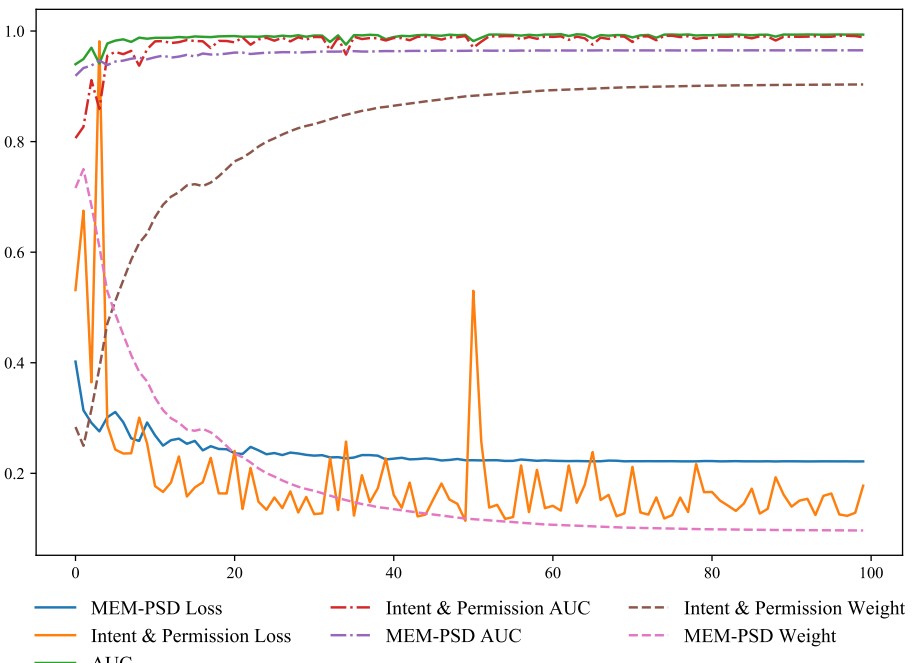

**Figure 15.** Adaptive Soft Voting ($\mu = 0$).

## 5. Conclusions and Future Work

In this paper, we propose a fast Android malware detection method. We propose a multidimensional feature engineering of Android application packages combining information entropy, file headers, and manifest files, propose adaptive shrinkage convolution to improve the convolution unit and propose a adaptive soft voting ensemble learning method, which enables efficient and accurate Android malware detection and provides a new idea for static Android malware detection. With the increasing camouflage and obfuscation techniques, our future research will further improve the detection accuracy and discrimination against unknown malware by combining dynamic analysis techniques.

We analyzed the performance indexes of our proposed several feature selection and extraction methods and base models by the above experimental results, and the performance index of our method is higher than that of single feature selection and extraction algorithm, which indirectly proves that single feature selection and extraction algorithm will miss some features, and in our future work, we will continue to find more expressive and less computational feature extraction schemes, and also strengthen the denoising capability of our model to achieve more efficient and accurate Android malware detection.

**Author Contributions:** Conceptualization, B.H.; Data curation, J.L.; Formal analysis, B.H. In addition, J.H.; Funding acquisition, B.H. In addition, J.L.; Investigation, B.H.; Methodology, T.P. In addition, B.H.; Project administration, T.P. In addition, X.H.; Resources, T.P., B.H. In addition, Z.Z.; Supervision, T.P. In addition, R.H.; Validation, T.P. In addition, B.H.; Visualization, T.P., B.H. In addition, J.L.; Writing—original draft, B.H.; Writing—review & editing, T.P. In addition, B.H. All authors have read and agreed to the published version of the manuscript.

**Funding:** This work is supported in part by Department of Education of the Hubei Province of China under Grant No. 2020BAB116 and No. Q20131608, Hubei Provincial Engineering Research Center for Intelligent Textile and Fashion and Engineering Research Center of Hubei Province for Clothing Information.

**Institutional Review Board Statement:** Not applicable.

**Informed Consent Statement:** Not applicable.

**Data Availability Statement:** Three publicly available datasets were used in this experiment, the CICMalDroid2020 and CIC-InvesAndMal2019 from the Canadian Institute for Cybersecurity Research (CIC), and the Drebin from the Institute for Systems Security at the Technical University of Braunschweig.

**Acknowledgments:** The authors extend their appreciation to the Department of Education of the Hubei Province of China for funding this research work through grant No.2020BAB116 and No.Q20131608. The help of Hubei Provincial Engineering Research Center for Intelligent Textile and Fashion and Engineering Research Center of Hubei Province for Clothing Information in this research completion is greatly appreciated.

**Conflicts of Interest:** The authors declare no conflict of interest.

## Abbreviations

The following abbreviations are used in this manuscript:

| | |
|---|---|
| MEM-PSD | Power spectral density calculated by maximum entropy method |
| AdaSV | Adaptive soft voting |
| ASCNN | Adaptive shrinkage convolutional neural network |
| MLP | Multilayer perceptron |
| H | Dex head feature |
| M | Power spectral density feature of entropy sequence of the Dex file |
| I | APK's permission and intent feature |
| C | Combined features of Dex head features and power spectral density features of entropy sequences of Dex files |

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
