# Peer review of "A Lightweight Multi-Source Fast Android Malware Detection Model"

_applsci, doi:10.3390/app12115394_

Round 1

Reviewer 1 Report

A lightweight multi-source fast Android malware detection model

The authors presented an interesting topic to detect malwares in Android devices in real-time is to be appreciated. The authors presented the paper in a splendid manner by covering all possible aspects of evaluation parameters. However, description of the algorithm is essential part and is major missing in the manuscript and a few corrections are listed to incorporate:

  • Typo and grammatical corrections to be taken care by the authors. Eg:             In abstractà line no.9: “thus improve the performance” to be modified as “thus improves the performance”. Line no. 68 “will choose to download applications from some third-party websites, these” not properly ended. Check the sentence in line no 74,
  • Authors shall discuss about using power spectral density of the structural entropy of the executable file in detail and size of the data considered for applying adaptive shrinkage convolutional neural network model.
  • Authors can elaborate in detail the need of using new adaptive soft voting method to reduce noise and jitter!! Without voting method, show the performance of the proposed algorithm.
  • Check the continuity between lines 146 and 147. Chk line 153 which is standalone.
  • Section 3 – Specific Approach name can be mentioned clearly rather than including generically.
  • All FIGURES must be cited inside text. A detailed explanation is expected for figure 1 and 2.
  • Section 3.1 heading “1. Android application structure and its feature selection and feature extraction scheme” can be modified as “3.1. Android application structure and its feature selection and extraction scheme”
  • Change the caption of Figure 3. Wherever possible, very specific captions/titles/headers to be included.
  • Beneath Figure 3, Base models can be enlisted one by one. What kind of base model has been chosen? No information is provided reg base models.
  • Elaborate the Base Models used and the integrated learning model applied and then project the results. Without knowing the kind of model applied and the parameters related to it, evaluation is worthless.
  • Equation no to be included in line no 276, 293, etc.,
  • An understanding of the algorithms is much essential. Elaborate adaptive shrinkage convolutional neural network model in detail and how the function gets updated.
  • Sigmoid and softmax function can make the model (CNN) more susceptible to problems during training, so-called vanishing gradients problem. Hence avoid using sigmoid and softamax fn in the proposed work. CNN doesn’t prefer sigmoid and softmax.
  • Most of the modern CNN models prefer ReLU and its extensions like Leaky RELU / Maxout. Kindly check and update the functions.
  • In table 1, cite the reference number against the method name. Provide a name to your proposed approach and replace “Ours” in table 1 and in section 3.

As an overall conclusion, I strongly recommend the manuscript to consider for Publication after incorporating the suggestions.

Decision: Major Revision

Reviewer 2 Report

The paper studies the “Android malware detection model” and claim to present a lightweight multi-source model. The authors have chosen an interesting topic to study but the paper exists some flaws as follows.

1- Motivation: The authors have written this section in a very general way and it is necessary to make corrections and motivation to be written exactly in accordance with the paper and its innovation.

2-  The authors claim that: “We propose an adaptive shrinkage convolutional neural network, which can dynamically adjust the convolutional kernel weights and activation function thresholds through an attention mechanism, making the convolutional network with denoising ability while improving the expressiveness of the neural network model.” There is no evidence to suggest a dynamic approach. Please explain.

3- Related Work: There has been a lot of research on static analysis methods, especially in Android datasets such as Drebin, Genome, and Contagio, which the authors did not mention either methods or datasets. Please explain.

4- Related Work: The authors have not described hybrid methods. Please explain.

5 - Soft Voting needs to be clearly explained.

6- The description of the proposed model in Figure 3 is not clear. It needs to be explained in full detail.

7- The time complexity of the proposed method is very high, the authors need to explain in this regard.

8- It is necessary to consider legend for Figures. Some of them, such as Figure 4, are not understandable. Also, all shapes need to be labeled on the coordinate axes.

9- Formulas need to be numbered and all numbers referenced in the text.

10- The proposed method needs to be written as a pseudo-code. This will help the reader to understand the proposed method.

11- The comparisons provided are interesting, but the values need to be compared for all machine learning metrics, including FPR.

12- Although the authors have referred to a small number of new articles, most are old references. With a simple search, a large number of new articles on Android malware have been published between 2020 and 2022, which require authors to thoroughly review them.

13- The authors are required to provide the referees with the implemented code of the paper at an URL.

14-  The English writing of the paper is poor and needs to be revised.

Reviewer 3 Report

The manuscript entitled “lightweight multi-source fast Android malware detection model” is written well.

The main strength of this manuscript is it attracts the attention of smartphone users toward privacy issues. The authors proposed a lightweight malware detection model developed using ensemble methods.

The manuscript is lacking behind due to some points that are mentioned below:

  1. Add some papers that are published recently related to smartphone security like:
  • Mitigating adversarial evasion attacks of ransomware using ensemble learning
  • MFDroid: A Stacking Ensemble Learning Framework for Android Malware Detection
  • Android Malware Detection with Deep Learning using RNN from Opcode Sequences.
  • HybriDroid: an empirical analysis on effective malware detection model developed using ensemble methods
  1. In addition to that, compare the performance of the proposed model with existing frameworks present in the literature and also compare it with free anti-virus software and the results add to the manuscript.

Round 2

Reviewer 1 Report

All the comments/suggestions are incorporated in the current version of the paper and I agree to publish the paper in Applied Sciences.

This manuscript is a resubmission of an earlier submission. The following is a list of the peer review reports and author responses from that submission.